# Horses in Lithuania in the Late Roman–Medieval Period (3rd–14th C AD) Burial Sites: Updates on Size, Age and Dating

**DOI:** 10.3390/ani12121549

**Published:** 2022-06-15

**Authors:** Giedrė Piličiauskienė, Laurynas Kurila, Žilvinas Ežerinskis, Justina Šapolaitė, Andrius Garbaras, Aurelija Zagurskytė, Viktorija Micelicaitė

**Affiliations:** 1Department of Archaeology, Vilnius University, Universiteto St. 7, 01513 Vilnius, Lithuania; laurynas.kurila@istorija.lt (L.K.); zagurskyte@outlook.com (A.Z.); viktorijamicelicaite@gmail.com (V.M.); 2Center for Physical Sciences and Technology, Mass Spectrometry Laboratory, Saulėtekio Ave. 3, LT-10257 Vilnius, Lithuania; zilvinas.ezerinskis@ftmc.lt (Ž.E.); justina.sapolaite@ftmc.lt (J.Š.); andrius.garbaras@ftmc.lt (A.G.)

**Keywords:** horse, biometry, south-eastern Baltic, Migration period, Viking period, medieval

## Abstract

**Simple Summary:**

The tradition of burying horses in Lithuania was the longest-lasting custom in Europe, which has resulted in about 2000 known horse burials. This paper publishes the osteometric data and age of horses found in Lithuanian cemeteries and castles between the 3rd and 14th C AD and discusses the dynamics of horse body size in order to test previously suggested hypotheses regarding the relationship between large horse body size and its military use, possibly non-local breed, and high social status of the owner. Size analyses have shown that, in general, the horses in Lithuania in the 3rd–11th C AD were small compared to those in Central and Western Europe or Scandinavia. More significant changes can be observed in the Middle Ages. In the 12–14th C AD, there was a much greater variety of horses and the expansion of taller (140–150 cm) individuals. However, the continued abundance of small horses in Medieval times, found buried with saddles and other equipment, allows one to renew the debate on the formation of the cavalry, the tactics of combat, and the social composition of horsemen in Lithuania.

**Abstract:**

The tradition of burying horses in Lithuania lasted from the Early Roman period until the late 14th C AD. It was the longest-lasting custom in Europe, which has left about 2000 known horse burials. This paper publishes the osteometric data and age of horses found in Lithuanian cemeteries and castles of the 3rd–14th C AD, over 200 individuals in total. These are the remains of all the horses still stored in Lithuanian institutions. The paper discusses the dynamics of horse body size in order to test previously suggested hypotheses regarding the relationship between large horse body size and its military use, possibly non-local breed, and high social status of the owner. Moreover, we are publishing the AMS ^14^C dates of 13 horses previously assigned to the Migration period. The research results corrected the existing chronology. The abundant data also allowed an assessment of the development of the size and age of the horses in Lithuania between the 3rd and 14th C AD. Osteometric analyses have shown that Late Roman–post-Migration-period horses were unusually large compared to the Viking and medieval horses in Lithuania. Meanwhile, we suggest that the semi-slender-legged 118–125-cm-tall horse, which predominated in the Viking period, is the most consistent with the local horse type. In general, the horses in Lithuania in the 3rd–11th C AD were small compared to those in Central and Western Europe or Scandinavia. More significant changes can be observed in the Middle Ages. In the 12–14th C AD, there was a much greater variety of horses and the expansion of taller (140–150 cm) individuals. However, the continued abundance of small horses in the medieval times, found buried with saddles and other equipment, allows one to renew the debate on the formation of the cavalry, the tactics of combat, and the social composition of horsemen in Lithuania.

## 1. Introduction

The first horse remains in Lithuania were discovered in the Šventoji 43 Subneolithic settlement and date back to 3900–3700 cal BC [1]. Horse bone fragments are found in settlements throughout the Stone and Bronze Ages [2], and the first horse graves in Lithuania date back to the Early Roman period [3]. The tradition of burying/sacrificing horses continued uninterruptedly until the Christianization of the Grand Duchy of Lithuania in 1387, and, as late as 1382, Kęstutis, the Grand Duke of Lithuania, was cremated together with his horses, dogs, and birds of prey [4]. Horses still occupy a unique place in Lithuanian memory and folklore [5,6], and the image of an armed rider (Vytis) has been the coat of arms of the state at least since the late 14th C AD.

Lithuanian archaeologists mostly concentrate their studies on horse burial customs and riding equipment deposited as grave goods. The topic of the Lithuanian war horse, however, has rarely been approached from the zooarchaeological perspective. Hypotheses have been suggested regarding the relationship between large horse body size and its military use, possibly non-local breed, and high social status of the owner [3,7], but these hypotheses are based on data from limited areas or chronological ranges. We analyze a large set of horse remains from Lithuanian burial and settlement sites in order to discuss the dynamics of horse body size throughout over a millennium, with an effort to test the above-mentioned hypotheses and to inspire new discussions on the use of the horse in Lithuania in the past.

The tradition of burying a horse alongside a human, either the whole animal or just parts of it, has been widespread in Europe and Asia since pre-Christian times [8,9]. Numerous works have been devoted to horses, their research methodology, size, burial customs, etc. (e.g., Refs. [10,11,12,13,14]). In Europe, the tradition mainly declined with the spread of Christianity, while, in Siberia, Central Asia, or the Caucasus, horse sacrifice remained until the 19th and 20th C AD [15,16]. Thus, on the one hand, the tradition of horse burial in Lithuania was not exceptional. On the other hand, it was the longest-lasting custom in Europe, which has left entire burial grounds where about 2000 horse graves have been studied to date. Considering the area of Lithuania (65,300 km^2^), this is a significant number.

Horses were buried in several ways: in burial mounds and cemeteries, in graves together with or apart from humans, often in separate parts of the same cemeteries, and in other ways. Horses were buried in their entirety or only in parts, such as the head or lower limbs, and there are also graves of cremated horses. The horse was the closest animal to humans in burial contexts and the only animal sometimes buried in the same manner as a human. The number of horses found in the burial sites also varies greatly, ranging from one horse in a burial mound to 200–300 individuals in the largest cemeteries of the Viking and later periods in central Lithuania. The traditions of horse burial in Lithuania have been discussed in numerous publications [7,17,18,19,20,21], but relatively little attention has been paid to the buried horses’ size and age.

During the 1400 years of the horse burial period, the horse and human graves dating back to the 4th to 6th C AD, i.e., the Migration period, were perhaps the most intriguing. Humans were often buried with numerous weapons and, often, non-local grave goods during this time. According to zooarchaeological studies carried out in the second half of the 20th C [22], horses of this period often reached a withers height of 133–144 cm, which is 15–25 cm taller compared to the Viking period and even medieval horses [7,23]. The exceptional morphology of the horses and the non-local grave goods have long been a source of debate about the non-local origins of the horses and riders. This led to hypotheses that the “large” Lithuanian horses may have arrived here together with the groups of warriors who took part in the Migration period battles and were possibly related to the large horses that spread in Europe after the fall of the Western Roman empire [12,24]. Moreover, both the archaeological material and the horse burial tradition in Lithuania itself point to at least two distinct directions of non-local influences. In eastern Lithuania, in the Migration period or slightly earlier, the graves of the warriors and horses buried in mounds have been associated with the Middle Danube region [18]. Only nine such burials have been investigated so far. In the burial sites of western and central Lithuania, people were buried with whole horses or only with horses’ heads and legs [3,25]. The latter tradition reached Lithuania from the west via the Sambia Peninsula (present-day Kaliningrad, Russian Federation), but it was quite widespread in Europe [8]. It had reached Scandinavia in the north [26], and its origins have been attributed to Asian nomads. About 150 such graves are known in Lithuania, where the tradition continued until the beginning of the second millennium AD.

However, it was clear that, in order to assess the morphological characteristics of the Migration period horses, it was necessary to carry out comparative studies on horses from this period as well as from later periods. Some of the Viking period–medieval horse skeletons had not been investigated before. Others were analyzed shortly after the excavations in the second half of the 20th C, and analysis data—brief information on the withers height and age of the horse—were presented in the archaeological reports. However, such data were unusable due to a few problems. Osteometric data on horse bones are not given in the reports; the methodologies used to determine height and age are usually not mentioned. It is clear that researchers used different methodologies (e.g., Refs. [22,27,28]). This has not allowed a reliable reconstruction of the dynamics of the size and age of the Lithuanian horses throughout the 3rd–14th C AD. Unfortunately, the possibilities of re-investigation are now limited by the fact that only a portion of the skeletons found in the burial sites have been stored in repositories so far.

To clarify the origins of the Migration period horses and humans, a project funded by the Research Council of Lithuania was carried out between 2019 and 2021. During it, ^87^Sr/^86^Sr stable isotope analyses of 13 horses and 40 humans, dated to the Migration period, were conducted. The horse and human remains were also AMS ^14^C-dated. The project also included the re-analysis of the size and age of all the horses available for investigation. The results of the ^87^Sr/^86^Sr analysis demonstrated that 3/13 (23%) horses were of non-local origin. The hypothesis that the horses are of Southern or Central European origin has been rejected, but an intriguing hypothesis of origin from southern Sweden has been suggested. The osteometric analysis of the horses revealed that 2/3 non-local horses were the largest, 137–138-cm-high individuals [29]. Human studies have shown that 12/40, i.e., 30%, of the individuals are of non-local origin and that they could have originated in present-day Lithuania, the Sambia Peninsula, the eastern Baltic coast, north-eastern Poland, or southern Sweden [30,31]. Such results have opened up new discussions about possible migration directions in the 3rd to 7th C AD, different from Southern and Central European vectors, which had dominated archaeological debates until now.

This article publishes the AMS ^14^C dates of 13 horses previously assigned to the Migration period, as well as the osteometric data and age of horses found in Lithuanian cemeteries of the 3rd–14th C AD, over 200 individuals in total. The research results corrected the existing chronology of the burials, and the osteometric data enabled a new evaluation of whether the horses from the Migration period are indeed exceptional in size in the context of Lithuanian horses from the 3rd to 14th C AD. The abundant data also allowed an assessment of the development of the size and age of the horses in Lithuania between the 3rd and 14th C AD.

## 2. Material and Methods

### 2.1. Zooarchaeological Analysis

Almost 400 horse skeletal remains found in Lithuanian burial sites of the 3rd–14th C AD were analyzed (Figure 1); however, due to poor preservation, only over 200 individuals were suitable for further size and age analysis. The burial sites were divided into three chronological groups. A total of 21 individuals from 6 burial sites were assigned to the Late Roman–post-Migration period (3rd–7th C AD); 13 horses from this period were directly AMS ^14^C-dated. These were individuals selected for the ^87^Sr/^86^Sr analysis, the results of which are published in a separate publication [29]. In order to differentiate between horses (*Equus caballus*), donkeys (*Equus asinus*), and mules, the methodology described by Johnstone [12] was followed.

The remains of 293 individuals dating to the Viking period (9–11th C AD) were examined from 8 burial sites. Most of them, with the exception of 7 individuals, were found in the Marvelė burial site in central Lithuania. Assessment of the archaeological context and the grave goods allows to expand dating of some of the burials into the 12th C AD. Therefore, the formal end of the Viking period (1066) has not been taken into account, both because of the approximate dating of these burial sites and because of the historical and archaeological context: no major break in the history of Lithuania before the establishment of the state in the 13th C AD is observed.

Horses from three burial sites dating back to the 13–14th C AD have been assigned to the medieval period. In this case, some of the burials may actually date to the 12th C AD but are difficult to distinguish from the later ones. The medieval horse group also included bones of horses from two castles and one town of the 13–14th C AD: 25 metatarsals and metacarpals were selected from Kernavė medieval town, 9 from the Vilnius Lower castle, and 8 from Memelburg, the castle of the Teutonic Order, present-day Klaipėda. We have included horses from the settlement sites in order to give as objective a picture of medieval horses as possible, based not only on the individuals from ordinary burial sites but also from contexts related to the higher social strata (Vilnius, Kernavė) and the Order. In total, the remains of 92 individuals were examined from the medieval period. During this research, skeletal remains of all Lithuanian horses known to the authors from 3rd–14th C AD burial sites were examined.

Many of the horse skeletons were poorly preserved and unsuitable for osteometric analysis, so the number of individuals varied in terms of examination of their age and height. The age of the horses was estimated by tooth eruption [32], wear of the incisors, and the height of the premolars and molars [33]. Fusion time of the epiphyses [32] was also used for the age assessment. Withers height was estimated from long bone measurements using the corrected coefficients of Kiesewalter [34,35]. Horses were classified according to Brauner’s metacarpal and metatarsal slenderness index classification [36]. Measurements according to von den Driesh [37] were made using an osteometric board and an electronic caliper with an accuracy of 0.01 cm. Horse skeletons were sexed based on the presence of canine teeth in male horses over 4–5 years of age. However, in many cases, sex of the animal could not be determined due to missing bones. Long bone measurements, sex, and the age of each individual are given in Appendix A. To test for significance, we used a *t*-test. Analyzed horses’ remains are stored in the Zooarchaeological Repository of Vilnius University, Faculty of History. Horse from Varliškės is stored in Kernavė Museum, and horse skeletons of Paduobė-Šaltaliūnė III and Salakas are stored in National Museum of Lithuania.

### 2.2. Radiocarbon Dating

Radiocarbon dating was carried out at the Centre for Physical and Technological Sciences, Vilnius (Lithuania). For sampling, preference was given to the mandible or skull, from which a tooth was also taken for ^87^Sr/^86^Sr analysis. In some cases (Plinkaigalis 3A, 3B, Pagrybis 145, 157), metacarpal and metatarsal bones were sampled when cranium bones did not meet the collagen quality criteria. Bone collagen extraction was performed using the acid–alkaline–acid (AAA) procedure, followed by gelatinization [38]. Samples were treated with 0.5 M hydrochloric acid, 0.1 M sodium hydroxide, and 0.5 M hydrochloric acid. Gelatinization of bone collagen was carried out in pH 3 solution at 70 °C for 20 h. The gelatin solution was filtered using a cleaned Ezee filter and freeze-dried. The same collagen aliquot was used for the ^14^C, δ^13^C, and δ^15^N measurements. The stable carbon and nitrogen isotope ratios in bone collagen were measured using an elemental analyzer (Thermo FlashEA 1112) connected to an isotope ratio mass spectrometer (Thermo Finnigan Delta Plus Advantage). To ensure that the collagen was of adequate quality, four collagen quality parameters were also measured in the bone samples and used as indicators of collagen integrity [39,40,41]. Conventional ^14^C dates were calibrated with the OxCal 4.4.4 software using the IntCal20 curve [42]. All the ^14^C dates are reported within the 95.4% (2σ) confidence interval.

### 2.3. Site Description

#### 2.3.1. The Late Roman–post-Migration Period, 3rd–7th C AD

Paduobė-Šaltaliūnė III is a barrow cemetery in eastern Lithuania and is part of a huge array of about 50 burial sites with about 1700 barrows. Between 1987 and 2008, 18 barrows were excavated. In barrow 17, a burial of a male and a horse was found, which was plundered, most probably soon after internment. The remaining grave goods are typical for Central, Southern, or Western Europe and allow the grave to be relatively dated to the 5th C AD [18,43]. The horse was buried on its left side without any equipment (Figure 2A). Only the horse’s cranium remained from horse skeleton. According to the ^87^Sr/^86^Sr analysis, the horse is non-local, with a possible origin in southern Sweden [29].

Pagrybis cemetery in western Lithuania is dated to the 4–7th C AD, excavated between 1980 and 1982. A total of 217 human graves were found here, of which 35 males and 3 children were buried with horse body parts, including the head and lower limbs (Figure 2B) [44]. Up until now, the remains of 10 horses have been stored, and the ^87^Sr/^86^Sr ratios of 4 individuals (graves 104, 145, 157, and 207) have been analyzed; individual 207 was estimated as non-local, with the possible origin in southern Sweden [29]. Withers height was determined for 6 and age for 10 individuals.

Pavajuonis-Rėkučiai barrow cemetery in eastern Lithuania was excavated in 1994 and 1996. It was dated to the 4th–early 5th C AD. One horse was found together with a male in a robbed grave in barrow 6 [18,45]. According to the ^87^Sr/^86^Sr analysis, the horse buried in this barrow was local, as well as the female, who was buried in the adjacent barrow with rich grave goods, typical for the Late Roman–Early Migration period Gothic culture of the Black Sea littorals and Southern and Central Europe [30,46]. Age and withers height of the horse was determined.

Plinkaigalis is a cemetery in central Lithuania dated to the 4–7th C AD. It was excavated between 1977 and 1984, and 372 human graves were found there. The cemetery is extremely rich in grave goods. Some items are of a non-local origin and are associated with the Sambia Peninsula and southern Scandinavia [47,48,49]. Traces of violent trauma in human bones and embedded arrowheads typical of Asian nomads indicate conflicts and possible (im)migration. The cemetery contained burials of four horses buried separate from humans, of which two horses were buried in one grave, 3. Horse 2 was lying on its belly, the left front leg half-extended, the other legs folded under the bottom. Horses 3A and 3B were lying on their left side. Right front leg of horse 3A was half-extended, the other legs folded under the belly. All legs of horse 3B were stretched out. All horses were buried with bridle-bits, and also spirals in the tail. ^87^Sr/^86^Sr analyses were carried out on three horses (graves 2, 3A, and 3B), and horse 3B was identified as non-local. Horse skeleton from grave 1 is lost.

Taurapilis is a barrow cemetery in eastern Lithuania dated referring to the grave goods to the 5th C AD. It was excavated between 1970 and 1971. Four horses were found in barrows 1, 4, 5, and 6. The horses were buried together with rich and well-armed males. An extremely rich warrior, so-called “duke” grave in barrow 5, contained a set of luxurious grave goods, typical of various Barbarian regions of Southern, Central, and Northern Europe [50]. The horses were buried on their left (horses 4 and 5) and right side (horses 1 and 6), with almost stretched legs. Horses 4 and 5 were without any equipment; horse 6 was buried with a bridle-bit in its snout. Skeletal remains of horses 4, 5, and 6 contained long bones and some teeth; therefore, ^87^Sr/^86^Sr analysis was carried out for all of them. All horses appeared of local origin. Withers height was estimated for all four horses.

#### 2.3.2. The Viking Period, 8/9–11/12th C AD

Degsnė-Labotiškės barrow cemetery in eastern Lithuania was excavated in 1970. After the examination of 18 barrow mounds, human and horse inhumations and cremations were found. Barrows 2, 5, and 14 contained destroyed horse graves and barrow 1 contained horse cremation. Horses were found squeezed into pits, lying almost on their bellies, with their legs hardly folded underneath. Horse 2 was buried without any equipment, horse 5 with bridle-bit found under the snout, horse 14 with bridle-bit in the snout. The horse graves date back to the 10–12th C AD [51]. The ages of all three horses and the sizes of two horses were determined.

Jakšiškis is a barrow cemetery in eastern Lithuania. It was excavated between 2000 and 2007. In 2000, a horse with a bridle-bit in its snout was discovered in the center of barrow 1. The grave dates back to the 9–12th C AD [52]. The age and size of the horse were determined.

Kretuonai barrow cemetery in eastern Lithuania was excavated between 1976 and 1980 and in 1994. During the latter excavations, horse burial was found in barrow 1. It was found buried in the center of the barrow, in grave 3, and was equipped with a bridle-bit. Bronze spiral from the horse’s mane was found near the head, and a sickle laid next to the horse’s pelvis bones [53]. The size of the horse was determined.

Marvelė is a burial ground in central Lithuania dating to the 2nd–12th C AD and is the largest excavated burial site in the Baltic States. Between 1991 and 2007, 236 horse burials (about 300 individuals) and 1591 human graves were discovered, including rich burials of the Late Roman–Migration period warriors with abundant weapons and other equipment. In some horse graves, the remains of two, three, or even more individuals were found. Horses were buried individually in separate parts of the cemetery, usually with bridle-bits, bronze spirals, amber beads, and bells. Remains of saddles (stirrups, buckles) were found in some horse graves as well. Most of the horses are dated to the Viking period; however, a small group was assigned to the Late Roman–post-Migration period [54,55]. Partial burials of horse head and limbs, as well as burials of scattered horse remains, were also discovered. Up till now, skeletal remains of ca 300 horses are stored. Due to poor preservation, the heights of only 118 horses and the ages of 206 horses were determined. ^87^Sr/^86^Sr analysis of the Late Roman period horse 113 was performed and it was identified as local [29].

Salakas is a barrow cemetery in eastern Lithuania. It was excavated between 2000 and 2001. Disturbed horse grave was found in barrow 2 and was dated to the end of the 1st millennium AD [56].

Varliškės barrow cemetery in eastern Lithuania was investigated between 1998 and 1999. Then, 10 barrows were excavated, and 23 human cremations and horse inhumation was found. Horse in barrow 7 (3) was buried on its right side—belly, with a bridle, of which circle and prolonged bronze plates, pellet bells, and bits remained. Larger bell was found under the horse’s breast [57]. The horse was AMS-dated to the 8–9th C AD [58].

#### 2.3.3. The Medieval Period, 12/13–14th C AD

Obeliai cemetery was excavated between 1979 and 1983. A total of 190 inhumated and nine cremated human graves and 30 inhumated horse graves were found there. A dog was buried with a horse in grave 4. Some of the graves were disturbed. The preservation of horse skeletons was poor, and only seven individuals were collected during the excavations; however, the remains of 4 horses have been preserved so far. Horses 3, 4, and 5 were buried with bridles, saddles (two stirrups and one belt buckle usually remained), and bell in every grave. Horse 17 was buried with bridle, bronze spiral, and 2 amber beads [59]. Withers height was estimated for all four horses; age was estimated for two individuals.

Pakalniškiai cemetery was excavated between 1963 and 1974. It contained 22 human cremations and 262 horse inhumations. The preservation of the bones was very poor, and horse remains were only collected in cases where at least one well-preserved long bone or cranium was found. For this reason, only 73 horse remains were collected. Together with horses, remains of bridles, saddles, and bells were found [27]. The remains of 11 horses are stored until the present day, and all of them have been studied.

Masteikiai cemetery was excavated between 1993 and 1994 and 68 human inhumations and cremations, as well as 62 horse burials, were found. Most of the graves were disturbed, and bones were poorly preserved. Some of the graves (about 25) contained the whole horse, while others contained only parts of the skeleton—head and lower parts of the limbs. Horses were buried with bridles, bronze spirals, amber beads, and bells. At least three graves contained remains of dogs [60,61]. One of the dogs was dated to 1163–1265 cal AD [62].

Kernavė medieval town was a large trade and craft center and strongly fortified residence of the Grand Duke of Lithuania between the 13 and 14th C AD. The castle and town were attacked by the Teutonic Order, burned down in 1390, and were never rebuilt. During the excavations of 1999–2001 [63,64,65,66], 13 metacarpals and 12 metatarsals of horses were found and were included in this study. All material is dated to the 13–14th C AD.

Vilnius Lower castle was the central castle of the Grand Duke in the capital of the Grand Duchy of Lithuania since the early 14th to the mid-17th C AD. It was first built as a part of Vilnius defensive system, and, later, after several stages of reconstruction, it became the residential palace. It was abandoned after Muscovian attack in mid-17th C AD and completely demolished at the beginning of the 19th C AD. During the excavations of 1988–2014, five metacarpal and four metatarsal bones of horses were found in the layers dated to the 13th–mid-14th C AD and were included in this study [23].

Memelburg (present-day Klaipėda) castle is located in western Lithuania. It was built between 1252 and 1253 by the Livonian Order and in 1328 was passed over to the Teutonic Order, which made it the Order’s northern-most castle in Prussia. In 1257–1258, Memelburg was granted Lübeck city rights, but, at the end of the 13th century AD, further urban and economic development failed due to the wars with local Curonians, Samogitians, and Lithuanians [67]. Horse remains discussed in this paper were excavated in 2016 in the northern part of the castle, in the layers of late 13th–early 14th C AD. Three metacarpal and five metatarsal bones were found and analyzed [68,69].

## 3. Results

### 3.1. Withers Height

The size of horses for the Late Roman–post-Migration period was represented by sixteen individuals from six burial sites. The withers height of the horses varied from 125.4 cm to 137.7 cm. The average height of all the horses was 130.2 cm (Table 1). The smallest horses were from graves Pagrybis 207 and Taurapilis 5, with a height of 125–126 cm. However, the Taurapilis 5 horse was only 3–3.5 years old. The largest horses were from Paduobė-Šaltaliūnė III and Plinkaigalis 3B, with a height of 137–138 cm. In general, the largest horses were found in the Plinkaigalis cemetery.

The size of the Viking period horses was mainly based on individuals from one burial site, Marvelė (118/124 individuals). The average withers height of the horses from this period was 123.6 cm. The lowest and the tallest individuals were also found in Marvelė, measuring 112.3 cm and 141.0 cm, respectively (Table 2). Later, in the cemeteries of the 12–14th C AD, 24 individuals were found, with an average height of 127.2 cm. The lowest was a 108.2-cm-tall adult horse from Obeliai burial 4, while the largest horse was a 150.9-cm-tall individual from Pakalniškiai burial 192 (Table 2). The 13th and 14th C AD horses from the Vilnius Lower castle and Kernavė were slightly larger, with an average height of 128.1 cm and 128.2 cm, respectively. In general, the mean withers height of all the horses from the 12–14th C AD was 127.9 cm. The largest medieval horses in modern Lithuania were found in Memelburg, the castle of the Teutonic Order. Here, eight horses’ bones were measured, and a mean withers height of 132.8 cm was estimated (Table 3).

### 3.2. Robusticity

The robusticity of 13 horses dating from the 3rd to the 7th C AD was assessed. The slenderness index of the metacarpal bones ranged from 14.0 to 16.4, with a mean of 14.9 (Table 1 and Table 2). The most robust metacarpal bones were found in Pagrybis; horses 145 and 157 were 16.0 and 16.4, respectively. Horse 113 from Marvelė also showed a larger (15.9) robusticity. The horses with the slender metacarpals were those found in Pagrybis, graves 142 and 207. The Viking period and medieval horses had more slender metacarpals compared to the earlier period, with a mean index of 14.6 and 14.7, respectively. The greatest robusticity (mean 15.3) was identified for Order horses (Table 3). The slenderness index of the metatarsals for the 3rd–7th C AD horses was 11.6 and slightly increased to 11.7 in the Viking period, whereas the 12–14th C AD individuals demonstrate significantly more slender metatarsals (*p* < 0.01).

### 3.3. Age

Age was determined for 18 individuals of the Late Roman–post-Migration period. Horses between 1–1.5 and 10 years of age were buried during this time (Table 1). The same number of young horses, aged 1–5 years and older, aged 5–10 years old, were found. The 7–10-year-old individuals were buried most often. In the Viking period, the age of the horses was more varied, with both foals under a year and horses over 15 years of age being buried. Horses aged 5–10 years were buried the most often (42%) during this time. During the Viking period, 7.1% of the horses (15 individuals) were younger than 2 years of age, of which 6 were younger than one year old; all of them were found in Marvelė. However, only one yearling foal was found in the medieval burial sites (Masteikiai 28), with the most abundant group of horses aged 5–10 years. There was an increase in the number of older horses during this period, with 21.3% of the horses aged 11–15 years, and 12.7% of the horses older than 15 years (Table 2).

## 4. Discussion

### 4.1. AMS ^14^C Dating of the Migration Period Horses

The 13 new horse AMS ^14^C dates from six burial sites (Figure 1 and Figure 3, Table 1), as well as 33 new human dates from this period [31], have extended the chronology of the burial sites presumed to date to the Migration period (late 4–6th C AD) based on the archaeological finds. Radiocarbon dating expanded the study beyond the expected limits to the 3rd–7th C AD, i.e., to the Late Roman–post-Migration periods.

The data obtained allow us to discern two or, which is even more likely, three stages of horse burial. As it turns out, Marvelė 113 horse (134–408 cal AD) comes from the earliest stage, the Roman period horse burial tradition, and is yet a rare example in central Lithuania. All the burials from eastern Lithuania, those from Paduobė-Šaltaliūnė III, Pavajuonis-Rėkučiai, and Taurapilis, structure a chronologically rather uniform group dating to a similar or slightly later period (ca 250–530 cal AD), i.e., the Late Roman or Migration period. More precise dates are hampered by the plateau in the calibration curve around 1700 and 1600 BP. It is, therefore, difficult to say whether these graves represent a relatively short episode or a longer tradition of horse burials. The former hypothesis would be indirectly supported by the small number of such graves. On the other hand, there is also no reason to try to generalize all the aforementioned graves as homogeneous. Eastern and central Lithuania do not share a common material culture, nor the common custom of burying horses itself. Therefore, the burial from Marvelė and those from eastern Lithuania may be representatives of two different episodes of horse burial tradition.

The Pagrybis horse sacrifices and Plinkaigalis horse burials, dating from around 550 to 650 cal AD, are later than previously thought; they fit the post-Migration period. Pagrybis grave goods and burial customs in particular have demonstrated that the dating of some cultural elements (e.g., partial horse burial) is later than the events of the Great Migration that swept across Europe a bit earlier. Thus, in the search for the origins of the nomadic tradition of horse sacrifice, the previously more attractive version of the Huns as its promoters might have to be reconsidered, possibly linking horse sacrifices to Avar influence. However, these few graves do not allow us to make firm generalizations either. It is possible that a late stage of an older local tradition has been encountered here. Burials of horse body parts in western Lithuania are known to have occurred since the Early Roman period [2,3]. However, since we do not yet have any radiocarbon dates from the intermediate period, the question remains open for the future of whether the Pagrybis burials were the continuation of the Roman period partial horse burial tradition or the horse sacrifice custom was resumed after a several-centuries-long break. In general, the number of horse graves from 3rd–7th C AD in western Lithuania is incomparably higher than in eastern Lithuania, which may indicate a continuous custom of horse sacrifice.

### 4.2. Dynamics of Withers Height

The study of horses from the 3rd to 14th C AD demonstrates that the largest horses in Lithuania (except those of the Teutonic Order) are found in the burial sites of the 3rd–7th C AD, i.e., in the Late Roman–post-Migration period, where the average withers height of the horses was 130.2 cm (Figure 4). The diversity in the horses’ height was the lowest (SD = 3.7), and no horses shorter than 125 cm were found. The withers height of 81.3% of the individuals was 125.5–133 cm, and two individuals were 137–138 cm tall (Figure 5). However, in the Viking period, the average height of the horses decreased to 123.6 cm, and 65.1% of the individuals were ranging from 108 to 125 cm high. Larger horses, 126–135 cm high, accounted for 33.1% of all the individuals, and this is the first time horses larger than 140 cm appear in Lithuania. However, despite the largest sample, the variation in the Viking period horses’ withers height remains low (SD = 5.1). In general, the Viking period horses were the smallest throughout the 3rd–14th C AD in Lithuania, and the difference in height between them and the horses of the 3rd–7th and 12–14th C AD is statistically significant (*p* < 0.001).

In the Middle Ages, the average height of the horses increases and reaches 127.9 cm (Figure 4 and Figure 5). Moreover, there is a significant increase in the diversity of the horses’ withers height during this time: the SD for the medieval horses is twice as high as before (SD = 10.2). The withers height of the horses found in the graves varied the most (SD = 11.6), with almost 43 cm separating the smallest (108.2 cm) and the largest (150.9 cm) horses. Horses shorter than 125 cm make up 45.5%, while horses larger than 140 cm were found in 13.8% of the graves, including large individuals of 151 cm and 156 cm. There was no significant difference in the size of the horses found in the burial sites, the Vilnius Lower castle, and the Kernavė medieval town (Table 2, Figure 4 and Figure 5). Although the average height of medieval horses is 2.3 cm lower than in the 3rd–7th C AD, the difference is statistically insignificant (*p* = 0.2). The tallest horses, as would be expected, were found in the castle of the Teutonic Order in the late 13th–early 14th C AD. The mean height of the horses here was 132.8 cm. However, the size of the Order’s horses was significantly different only from the Viking period individuals (*p* < 0.001). In addition, the horses from Order castle show the highest SD (13.0) of withers height.

Before assessing the size of horses from one or another period, we will attempt to describe the size of the local type of horse that might have been common in these areas at least until the Middle Ages. The earliest horse in Lithuania for which the withers height (122.3 cm) was estimated was found in the Aukštkiemiai cemetery in western Lithuania and is dated to 26–210 cal AD., i.e., the Early Roman period [2,3]. It is unclear what size the horses were in the settlement sites in the first millennium AD as we do not have well-preserved horse bones from this period. The horses of the 3rd to 7th C AD in Lithuania are unusually tall (see further), and ^87^Sr/^86^Sr studies have shown that at least 23% of them are of non-local origin [29]. Therefore, we did not use horses from this group as a basis for the reconstruction of the local horse type. It is likely that the Viking period material, with horses predominantly smaller than 123–125 cm, is the best representation of the local horses. Local horses of similar size were probably present in earlier times. This assumption would be confirmed by the small horses of the Germanic and other Barbarian parts of the world before the large Roman horses spread to those territories. The withers height of the Germanic and Gaul horses in the pre-Roman period was about 123 cm, which is very close to the height of the horses in Lithuania in the Viking and probably in the Roman period. According to the Romans, the horses of the Germanic and Gaul tribes were small and ugly, although strong and hardy [12]. The Order’s written sources describe the Baltic horses in a very similar way. In fact, most of them refer to neighboring Prussian autochthones. According to the Order, the local horses here were shaggy, small, 116 cm or slightly larger, fast, hardy, and universal, used for riding, work, and war (see Ref. [70] and the authors cited therein). The horses in Lithuania were also described as small and slender-legged by foreign travelers who visited the eastern and south-eastern Baltic region in the medieval and Early Modern period [71,72]. Small horses, averaging 120–123 cm in size, are common in Lithuania during the Early Modern period, up until the 20th C AD [23,73]. The zooarchaeological, historical, and neighboring material suggests that, in general, the local Lithuanian horse before the 12–14th C AD was a slender-legged or semi-slender-legged individual of about 118–125 cm in height. Therefore, horses taller than 130 cm are very likely to be of non-local origin or descendants of non-local horses. The native Lithuanian Žemaitukai horse breed that some researchers directly relate to horses of the 5–6th C AD are slender-legged, short-necked, pony-type horses 131–136 cm tall [74]. However, only DNA analysis may answer to what extent they are related to the horses of the periods we are discussing.

Despite 3rd–7th C AD Lithuanian horses with an average withers height of 130.2 cm being small in the European context [10,12,75,76,77,78], in the local 3rd–14th C AD context, however, these horses appear large. The difference is statistically significant (*p* < 0.001) when comparing horses from the 3rd–7th C AD and the Viking period. It is still difficult to explain this difference and the exceptional size of the Late Roman–post-Migration-period horses. It can be noted that, for a variety of reasons—trade, looting during the Roman and Migration periods—horse height increases in areas under Roman influence, e.g., Britain, and also in the Barbarian world [12,76,77]. Changes in horse height and the appearance of larger-than-normal horses are traditionally explained by the non-local origin of horses [79,80,81]. Our primary hypothesis was that the large Lithuanian horses of the Late Roman–post-Migration periods, or at least some of them, are of non-local origin but originated from the former Roman Empire or its peripheries. The analysis of ^87^Sr/^86^Sr of Lithuanian horses partially supported this hypothesis. A total of 3/13 non-local horses were identified, of which 2/3 were the largest individuals. However, the area of origin of the non-local horses could be southern Sweden instead of the expected Roman provinces. Non-locals were the horses from Paduobė-Šaltaliūnė III (138 cm), Plinkaigalis 3B (137 cm), and the smallest (125 cm) horse of this period (Pagrybis 207). Despite the fact that most of the 3rd–7th C AD horses were locals according to the ^87^Sr/^86^Sr analysis, we cannot reject that those other horses larger than the typical local ones could be the second or later generation descendants of non-local horses, or that they originated from foreign regions with a ^87^Sr/^86^Sr similar to Lithuania [29], or even demonstrate the attempt of horse improvement with a new breeding stock. Osteometrical data do not contradict the hypothesis regarding the origin of the three horses mentioned above from southern Sweden, while archaeological data can only partially support it. In addition, ^87^Sr/^86^Sr studies of humans from the same period have shown that 30% of the studied people were non-locals. Their place of origin may have been Sambia and southern Sweden as well, but no individuals with a southern or Central European origin were found [30]. Thus, it is possible that some of the horses may have originated in Sambia, where horses of similar size and build were bred during the Roman period [82]. Given the intense migratory processes during this period, horse mobility may also have been higher than usual and resulted in a higher number of non-local horses in the Late Roman–post-Migration period.

However, the new ^14^C dates suggest that many of the horses, previously dated to the Migration period, are actually younger and come from the late 6–7th C AD, when the Avars settled in Europe and the traditions and goods associated with them, including horses, began to spread. Perhaps the Avar stirrups reached Lithuania at a similar time [21] (pp. 191–197). Avar horses averaged 135–136 cm in height and have rather robust metacarpal bones [10,14,83]. Thus, the primary hypothesis that unusually large horses spread in the territory of Lithuania after the fall of the Western Roman Empire and may be related to the Romans or the Huns should be corrected. The horse graves dating back to the 6–7th C AD may already be related to the Avars or their direct or indirect influence. Some horses from the 6–7th C AD (e.g., Pagrybis 145, 157) had unusually robust metacarpals for Lithuania, with indexes of 15.9–16.4. This index of metacarpal slenderness is typical for Eastern nomadic horses [10]. However, it must be kept in mind that the physical characteristics and origins of the horses should not be directly linked to archaeological features. Customs, items that later became grave goods, and the individuals themselves (both humans and horses) may have spread in different directions, reaching, crossing, and becoming embedded in different cultural environments. The best example of this in Lithuania is the above-mentioned custom of burying horse body parts. Originating in western Lithuania around the turn of the eras [2,3], it has survived several waves of nomadic influences in Europe and possibly several episodes of immigration to Lithuania. Therefore, it is very difficult to answer whether the non-local horses that reached Lithuania in the 4th to 7th C AD were buried in accordance with non-local customs or in accordance with a long-lasting local tradition that was once established due to completely different influences.

Pagrybis horses stood out in another detail. The craniums of four horses (graves 104, 157, 207, and the skull “from a fireplace”) had injuries on their frontal bones (Figure 2B and Figure 6). The skulls of the remaining horses were too fragmented to be examined. The perfectly preserved cranium of horse 104 clearly demonstrates a fracture of the frontal bone, possibly from the blunt end of the axe. A similar method of killing horses was widespread (e.g., Refs. [84,85,86,87]). However, these and the Viking period horse from Antasarė barrow cemetery [88] are the only cases in Lithuania known to the authors of this paper. It is likely that there were more, but the poorly preserved craniums do not allow us to judge how widespread this custom was in Lithuania.

Grave goods of Germanic origin for a long time suggested that the larger horses of the 3rd–7th C AD may have reached the south-eastern Baltic region from the west, possibly from Sweden, as was indicated by ^87^Sr/^86^Sr studies. However, the question that remains unanswered regarding the 3rd–7th C AD horses is: where are the “normal”, small, 116–125-cm-height horses that are typical for these regions and were abundant in the Viking period and later (Figure 5)?

We tend to consider the Viking period horses with a small variation in size (SD = 5.1) as the typical local horses. However, during the 12–14th C AD, there are notable changes, with a strong increase in the variety in the size of the horses. In the Middle Ages, two groups of horses emerge (Figure 5). In the Order’s castle and in the Lithuanian medieval material, there are small horses, usually up to 122–123 cm tall, and individuals usually higher than 134–135 cm; e.g., in Memelburg, three individuals were 116–123 cm tall, and four individuals were 139–149 cm tall. We assume that the small horses were locals, and the large ones were bred by the Order’s brothers. The size of the large horses from Memelburg corresponds to the size of the horses found in the Order’s castle in Cēsis (Latvia), and also in the Order’s castles in Poland, as well as in Viljandi castle in Estonia [85,89,90]. Different types of horses are also found in medieval Poland. Some are small, 120–130 cm slender horses; others are larger, 130–140 cm tall. Among the smallest horses in medieval Poland, averaging 127–129 cm in height, were individuals from Prussia, the territory of the Western Balts. The tallest horses, with an average height of 134.5 cm, were bred in western Poland and can be related both to the influence of western Europe and to the legacy of the Teutonic Order, who settled in the territory of present-day Poland [80,91]. A similar situation in the 12–14th C AD can be seen in present-day Latvia [89], as well as in Lithuania. Here, the spread of large horses in the 13–14th C AD could, in many cases, be linked to the appearance of the Order in the south-eastern and eastern Baltic region in the early 13th C AD. Lithuanians obtained large horses in various ways: by capturing herds from the territories under the Order’s control, by keeping them as spoils of battle, by buying them, and sometimes by receiving them as gifts [23,75,92].

According to Makowiecki [80], the function of large and small horses was different: small local horses were used as pack and draught animals, while 130–140 cm horses were used by warriors as riding or warhorses. However, at least in the case of Lithuanian and Prussian horses, we would disagree with this view. The small native horses were undoubtedly also used as riding horses or even warhorses. This is also evidenced by the small horses with a height of only 108–115 cm, which were buried in medieval cemeteries up to the end of the 14th C AD (Table 2). They were buried with bridles, decorations, and sometimes saddles. For example, the smallest medieval horse (Obeliai 4, 108.2 cm) was found in the grave with the remains of a bridle, a bell, and a saddle, as was the case with most of the horses buried in this burial site [93]. Of course, small local horses were also used as pack animals. They were not suitable for the Order’s cavalry in the Baltic, but they were bred in Prussia and accompanied the Order’s troops on campaigns. Moreover, the Order’s written sources mention that Prussian nobles served in the Order’s ranks as lightly armed warriors, mounted on their own small local horses [75]. This shows that, even when given a choice, local warriors chose to use small local horses. This is in line with Irish sources, which point out that a good horse did not have to be large [94]. A recent study regarding the English medieval horses demonstrates similar results [95].

### 4.3. Robustness

Despite the differences, all the groups of horses we have identified fall into the same semi-slender-legged group according to the metacarpal index. The lowest metacarpal index (14.6) was found in the Viking period horses. Despite the increase in the height and size diversity, the metacarpal index (14.7) remains almost the same in the local medieval material, with the highest index (15.3) in Memelburg castle (Figure 7 and Figure 8). The Viking period horses are similar in the robusticity of metacarpals to the horses of Gauls and Germans of the pre-Roman period, which had a metacarpal index of about 14.3–14.8 (Figure 9). In the Roman period, larger horses (index 15.0–15.2) are found in these areas [12]. The average metacarpal index of Avar horses was 15.24 [10]. Small, but relatively robust (mean 16.02), horses were found in the Sambia peninsula during the Roman period [82]. In some cases, Sambian horses are similar to Lithuanian horses (Figure 9). Relations with Sambia are also evidenced by grave goods, found in human graves in Plinkaigalis, Pagrybis, and other sites [44]. The observed dynamics of the metacarpals index are similar to the changes recorded in the Barbarian world during the pre-Roman and Roman periods, which further suggests the significant influence of non-local horses on the local population. This suggests speculating that the south-eastern Baltic inhabitants in one way or another were involved in the tumultuous events in Europe during the Late Roman–post-Migration period.

According to the metatarsal robusticity index, all the horses analyzed fall back into one group of slender-legged horses. However, in this case, we observe different dynamics: although slightly, the robusticity of the metatarsals increases from 11.6 in the Late Roman–post-Migration period to 11.7 in the Viking period, while, from 12 to 14th C AD, we observe very slender-legged horses with an index of 11.1, similar to the Norman (1066–1200) horses in England. In Lithuania, a more significant increase in metatarsals robusticity can be observed in the Vilnius Lower castle material of the late 14th–early 16th C AD (Figure 7). In England, such a process can be observed as early as the 13–14th C AD and reflects a trend towards the development of an early type of heavy horse [95].

Unfortunately, we still have no plausible explanation for the different dynamics of robusticity observed between metacarpals and metatarsals. Perhaps we can tentatively speculate that the robusticity of the Viking period horses may reflect the specificity of the local horse type, while, in the medieval period, the influence of non-local horses with a different build may have become more prominent.

### 4.4. Age of Horses

From the Viking period onwards, older horses were buried, and this tradition was even more pronounced in the material of the 12–14th C AD (Figure 10). In the 3rd–7th C AD, no individuals over 10 years were found, whereas 22.2% and 34.1% individuals over 10 years were identified among the horses of the Viking period and the Middle Ages, respectively. The age structure of the 3rd–7th C AD horses differed statistically significantly from the later periods (*p* < 0.001). However, there was no significant difference between the horses from the Viking period and medieval burial sites (*p* > 0.05).

Horses of various ages have been buried in Europe; however, the most common were horses killed in their prime, 4–10 years [96]. In Europe, the tradition of horse burial ceased with the spread of Christianity around the 10th C AD. Meanwhile, Lithuania was only Christianized in 1387, and horses were still being buried in the 14th C AD. Burial of young horses suitable for riding and horseback combat may be the reflection of their importance in life. It comes as no surprise in the mid-1st millennium AD, the time of increased military activity. Later, seeing horses buried at an older age may indicate that the old tradition of burying horses had slowly faded over the centuries and lost its original ideas. Horse burial had become more practical, and, more often, the animals were older, less valuable, and even unsuitable for riding; e.g., a horse from the Kretuonai burial had far-advanced spavin of both hind legs, with metatarsal, tarsal bones, calcaneus, and talus ossified (Figure 11). The horse was hardly suitable for riding. The older age of the horses is probably responsible for the very high number of cases of spavin in the graves of the 12–14th C AD. A total of 13 metatarsals were found in medieval graves, and five (38.5%) had a spavin (Table 2 and Appendix A). Usually, such pathologies develop in horses as a result of overriding and sometimes as a result of trauma [12]. In the early 20th C AD, veterinarians mentioned that a large number of horses in Lithuania suffer from spavin and that the farmers themselves are to blame for this as they start work with horses when they are only 1.5–2 years old [97]. However, we will not discuss horse health and pathologies in detail in this paper. It is a complicated topic also due to the poor preservation of skeletons and the different practices for collecting and storing the bones.

## 5. Conclusions

The study of the skeletal remains of 3rd–14th C AD horses and radiocarbon dating have corrected the view of the horses in Lithuania during the first millennium AD. Radiocarbon dating has extended the chronology of the burial sites from the Migration period to the 3rd–7th C AD. The search for links with the Roman, Germanic, or Hunnic world can now be supplemented by assumptions regarding links with the Avars and the area of their influence.

The study has corrected the image of the Lithuanian war horse in the 1st millennium AD suggested by earlier researchers. Osteometric analyses have shown that horses in Lithuania in the 3rd–7th C AD were not as tall as previously thought, with an average height of 130.2 cm, and some individuals were identified as shorter, by almost 20 cm. This encourages one to reassess the role of the horse in the contemporary warfare and its relation to the military elite. However, the horses from the Late Roman–post-Migration period remain unusually large compared to the Viking and medieval horses in Lithuania. The phenomenon of unusually large horses for this region is partly explained by the ^87^Sr/Sr^86^ analysis results, where non-local horses 137–138 cm tall were found in the graves of the 3rd–7th C AD. Meanwhile, we suggest that the semi-slender-legged 118–125-cm-tall horse, which predominated in the Viking period, is the most consistent with the local horse type. In general, the horses in Lithuania in the 3rd–11th C AD were small compared to those in Central and Western Europe or Scandinavia. More significant changes can be observed in the Middle Ages. In the 12–14th C AD, there was a much greater variety of horses and the expansion of taller (140 cm–150 cm) individuals. This is likely related to the arrival of the Teutonic Order in the eastern Baltic in the fourth decade of the 13th C AD and may also be related to the military campaigns of Lithuanians in the eastern Slavic lands between the 12 and 13th C AD. However, without DNA analysis, it is hard to explain the changes in horses’ size in the 1st millennium AD or to confirm possible links between the large horses of the 12–14th C AD and the horses of the Order.

The increased number of older horses buried, which is observed in the Middle Ages, can be related to various reasons. Such changes may reflect longer exploitation of horses in the 12–14th C AD. The reasons for this phenomenon could be very different, e.g., better keeping conditions and diet. On the other hand, the longer exploitation may also reflect a general shortage of horses. However, these hypotheses require detailed studies on the nutrition, pathologies, and the ages of the horses. In the Middle Ages, the same custom of burying horses with saddles and other equipment was practiced for both very small individuals, 108–115-cm-tall, and the largest, 140–150-cm-tall, horses. This indicates not only the great variety of horses ridden but also that the local warriors did not avoid using even the smallest local horses. These data also allow renewing the debate on the formation of the cavalry, the tactics of combat, and the social composition of horsemen in Lithuania.

## Figures and Tables

**Figure 1 animals-12-01549-f001:**
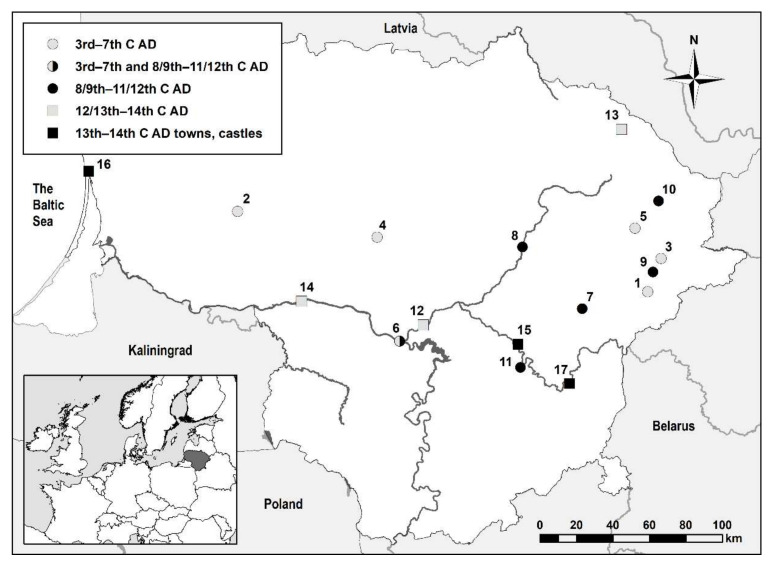
Location of sites mentioned in the text: 1—Paduobė-Šaltaliūnė III, 2—Pagrybis, 3—Pavajuonis-Rėkučiai, 4—Plinkaigalis, 5—Taurapilis, 6—Marvelė, 7—Degsnė-Labotiškės, 8—Jakšiškis, 9—Kretuonai, 10—Salakas, 11—Varliškės, 12—Obeliai, 13—Pakalniškiai, 14—Masteikiai, 15—Kernavė, 16—Memelburg (Klaipėda), 17—Vilnius.

**Figure 2 animals-12-01549-f002:**
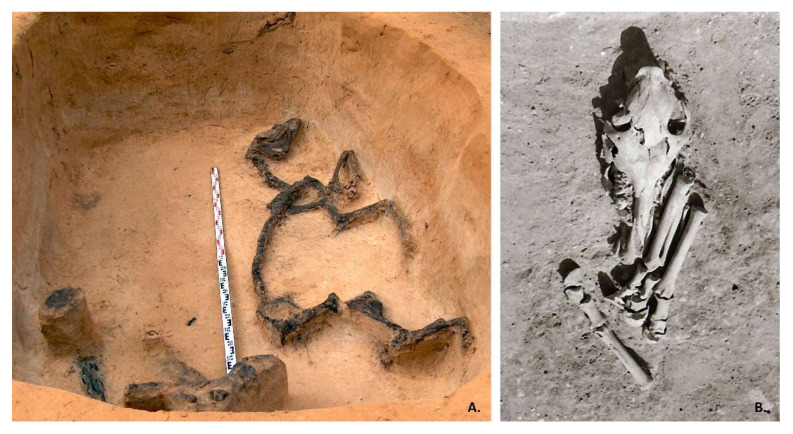
Human and horse burial in Paduobė-Šaltaliūnė III, photo by V. Steponaitis, 2006 (**A**); horse sacrifice in burial 207 in Pagrybis (**B**), photo by L. Vaitkunskienė, 1982, Archive of Lithuanian Institute of History, No. 51198.

**Figure 3 animals-12-01549-f003:**
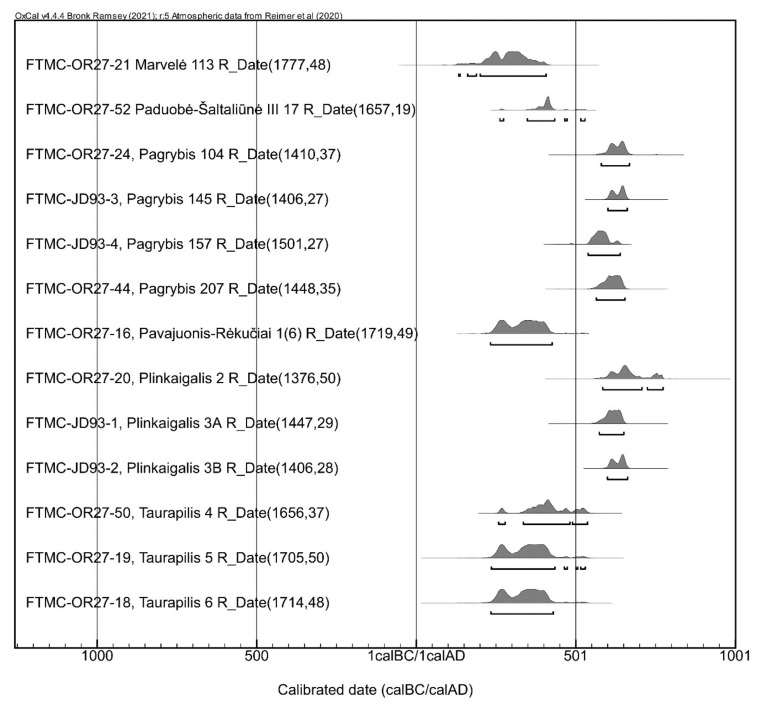
Calibrated ^14^C dates.

**Figure 4 animals-12-01549-f004:**
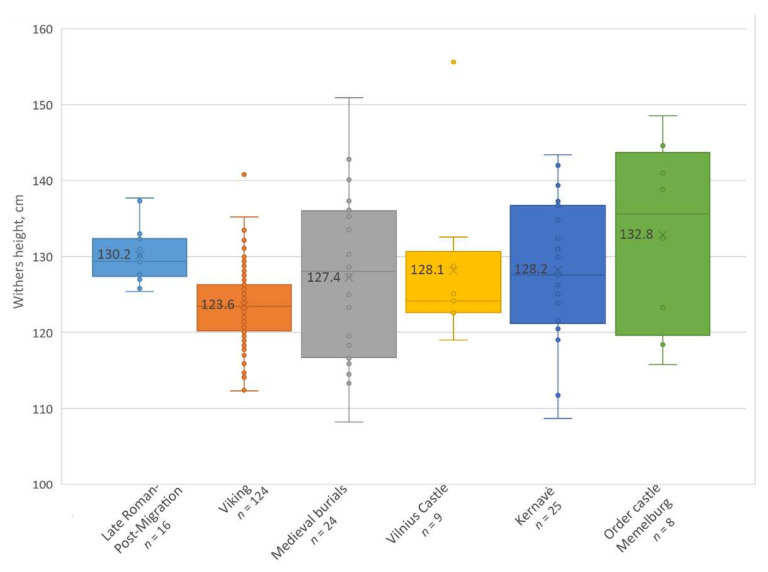
Boxplot showing withers height of horses during the Late Roman–medieval periods in Lithuania. Mean is written on the box and line demonstrates median.

**Figure 5 animals-12-01549-f005:**
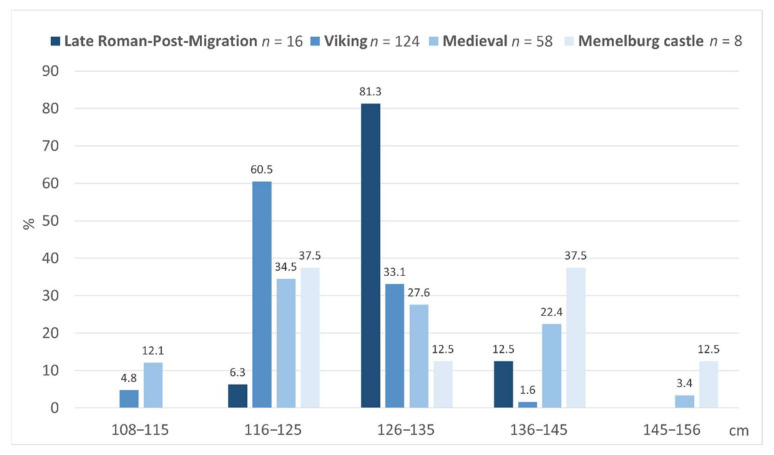
Withers height of horses.

**Figure 6 animals-12-01549-f006:**
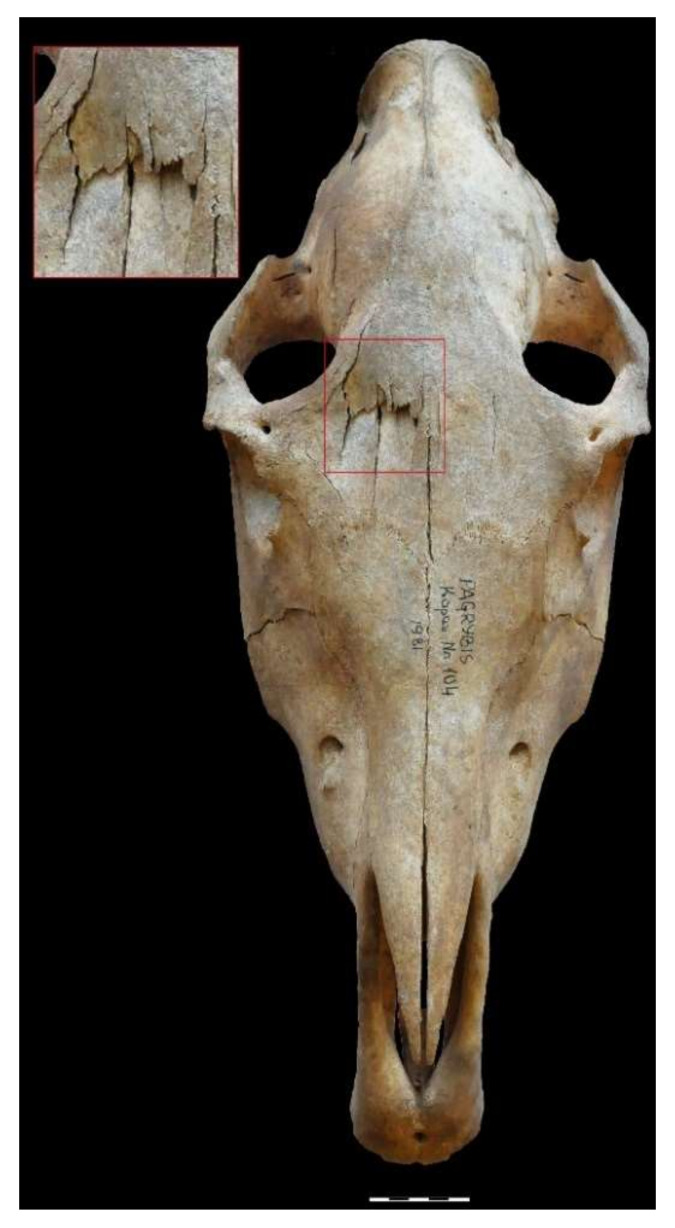
Pagrybis 104 horse male, 3.5–4 years old, with a fractured frontal bone. Individual had upper wolf teeth (P^1^), and its lower P_2_ teeth had traces of bit wear.

**Figure 7 animals-12-01549-f007:**
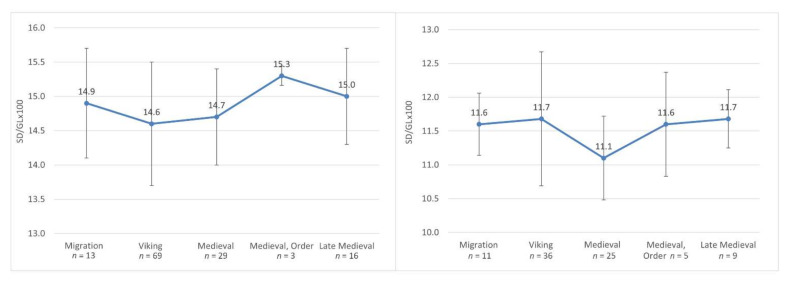
Mean SD/GL ratio through time for metacarpal and metatarsal bones. Bars are mean ± SD.

**Figure 8 animals-12-01549-f008:**
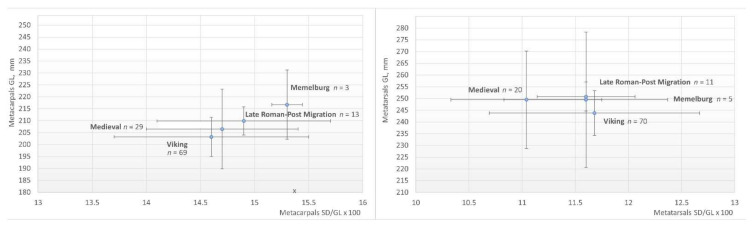
Scatter plot of mean ratios of measurements SD/GL × 100 of metacarpal bones (bars are mean ± SD).

**Figure 9 animals-12-01549-f009:**
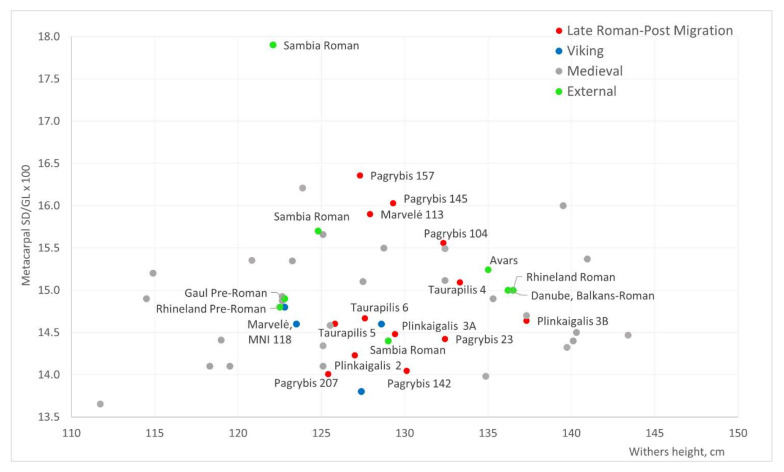
Withers height and metacarpal index (SD/GL × 100) of 3rd–14th C AD Lithuanian and horses from external regions.

**Figure 10 animals-12-01549-f010:**
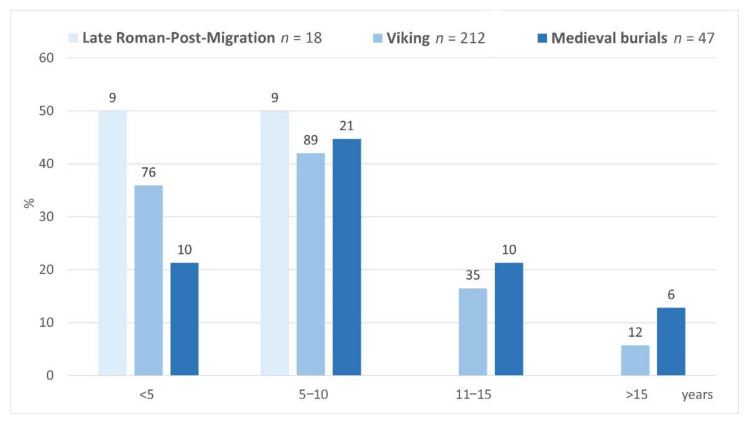
The age distribution of individuals. Above the columns is given number of individuals.

**Figure 11 animals-12-01549-f011:**
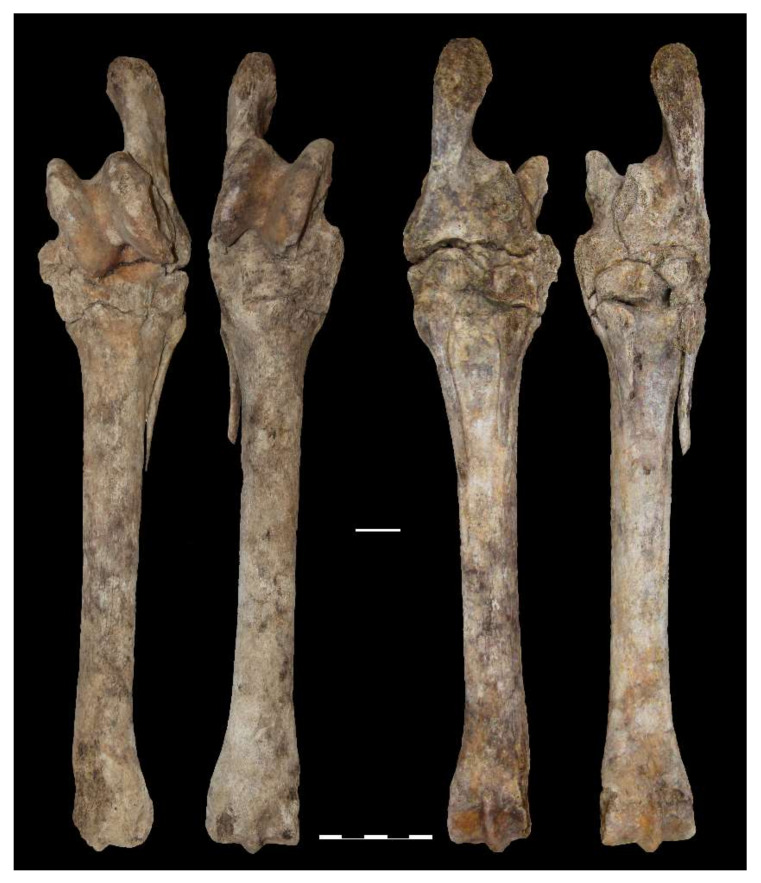
Kretuonai Viking period horse with spavin of both hind legs.

**Table 1 animals-12-01549-t001:** The Late Roman–post-Migration period horses studied. Reference for ^87^Sr/^86^Sr data Ref. [29]. * Bones whose epiphyses had not yet fully fused were also measured.

No.	Burial Site, Grave	Number of Measured Bones	WH Mean, cm	WH Mean/Site	MC SD/GL × 100	Age, Years	Lab. Code	AMS ^14^C BP	Cal AD 95.4%	%N	%C	C/N	% Collagen	Origin According to ^87^Sr/^86^Sr
1	Marvelė 113	6	127.9	127.9	15.9	9–10	FTMC-OR27-21	1777 ± 48	134–408	1.1	3.3	3.5	1.0	local
2	Paduobė-Šaltaliūnė III 17	1, cranium	137.7	137.7	-	7–8	FTMC-OR27-52(1)	1657 ± 36	259–537	9.2	29.0	3.1	5.7	non-local
							FTMC-OR27-52(2)	1667 ± 36	256–535	8.1	25.6	3.2	8.6	
							FTMC-OR27-52(3)	1652 ± 28	262–535	12.5	34.2	2.7	3.1	
							FTMC-OR27-52 (Oxcal R_Combine)	1657 ± 19	263–530	-	-	-	-	
3	Pagrybis 23	1	132.4	129.5	14.4	5–6	-	-	-	-	-	-	-	-
4	Pagrybis 55	-	-		-	1–2	-	-	-	-	-	-	-	-
5	Pagrybis 73	-	-		-	3–4	-	-	-	-	-	-	-	-
6	Pagrybis 92	-	-		-	2.5+	-	-	-	-	-	-	-	-
7	Pagrybis 99	-	-		-	4–5	-	-	-	-	-	-	-	-
8	Pagrybis 104	2	132.3		15.6	3.5–4	FTMC-OR27-24	1410 ± 37	580–668	10.2	27.9	3.2	4.9	local
9	Pagrybis 142	2	130.1		14.0	1.5+	-	-	-	-	-	-	-	-
10	Pagrybis 145	2	129.3		16.0	8–9	FTMC-JD93-3	1406 ± 27	601–662	15.7	45.0	3.3	3.3	local
11	Pagrybis 157	2	127.3		16.4	9–10	FTMC-JD93-4	1501 ± 27	540–640	15.5	44.4	3.3	3.3	local
12	Pagrybis 173	-	-		-	9–10	-	-	-	-	-	-	-	-
13	Pagrybis 207	2	125.4		14.0	7–8	FTMC-OR27-44	1448 ± 35	565–654	12.3	33.8	3.2	3.1	non-local
14	Pavajuonis-Rėkučiai 6	3	129.3	129.3	-	5–7	FTMC-OR27-16	1719 ± 49	234–426	4.7	12.9	3.2	2.6	local
15	Plinkaigalis 2	4	127.0	131.2	14.2	4–5	FTMC-OR27-20	1376 ± 50	585–774	2.0	5.7	3.3	1.5	local
16	Plinkaigalis 3A	1	129.4		14.5	1.5–2	FTMC-JD93-1	1447 ± 29	574–651	15.5	44.3	3.3		local
17	Plinkaiigalis 3B	2	137.3		14.6	7–8	FTMC-JD93-2	1406 ± 28	600–663	15.7	44.7	3.3		non-local
18	Taurapilis 1	3	131.0	129.4	-	3.5+	-	-	-	-	-	-	-	-
19	Taurapilis 4	5	133.3		15.1	8–10	FTMC-OR27-50	1656 ± 37	259–538	5.5	17.2	3.1	8.4	local
20	Taurapilis 5 *	3	125.8		14.6	3–3.5	FTMC-OR27-19	1705 ± 50	236–530	10.5	28.6	3.2	2.6	local
21	Taurapilis 6 *	4	127.6		14.7	3–3.5	FTMC-OR27-18	1714 ± 48	235–430	10.6	28.1	3.1	3.4	local

**Table 2 animals-12-01549-t002:** Horses from the Viking and medieval period burial sites.

No.	Period	Burial Site, Grave	Number of Measured Bones	Withers Height	Mean WH/Site	MC SD/GL × 100	Age, Years	Pathology
1	Viking	Degsnė-Labotiškės 2	5	128.6	125.7	11.6	5–6	
2		Degsnė-Labotiškės 5	5	122.8		14.8	15+	spavin
3		Degsnė-Labotiškės 14	-	-		-	1–1.5	
4		Jakšiškis 1	4	120.7	120.7	-	8–9	
5		Kretuonai 1	3	127.4	127.4	13.8	4.5+	spavin
6		Marvelė, MNI 118	174	123.5	123.5	14.6		
7		Salakas 2	2	128.5	128.5	-	1–1.5	
8		Varliškės 7	3	124.3	124.3	-	7–8	
10	Medieval	Masteikiai 11	1	135.0	129.3	-	8–9	
11		Masteikiai 40	1	128.8		-	1.5+	
12		Masteikiai 41	2	142.8		-	5–6	
13		Masteikiai 45	3	125.1		14.1	11–12	abnormal teeth wearing, inflammation of teeth roots
14		Masteikiai 46	3	139.5		16	9–10	
15		Masteikiai 48	1	113.0		-	8–9	inflammation of canine roots
16		Masteikiai 50	3	115.9		-	12–14	
17		Masteikiai 51	2	140.1		14.4	12–13	spavin, abnormal teeth wearing, inflammation of teeth roots
18		Masteikiai 52	2	123.9		-	12–13	spavin
19		Obeliai 3	1	137.3	124.8	-	1.8+	spavin
20		Obeliai 4	5	108.2		14.8	3.5+	
21		Obeliai 5	1	135.3		14.9	5–6	
22		Obeliai 17	4	118.3		14.1	3.5+	
23		Pakalniškiai 1 (163)	2	128.7	126.9	-	3.5+	
24		Pakalniškiai 2 (164)	3	114.5		14.9	7–8	
25		Pakalniškiai 20	2	127.5		15.1	7–8	spavin
26		Pakalniškiai 165	3	140.3		14.5	13–15	abnormal teeth wearing, inflammation of teeth roots
27		Pakalniškiai 167	4	136.1		-	8–9	
28		Pakalniškiai 175	1	130.3		-	8–9	
29		Pakalniškiai 184	3	119.5		14.1	3.5+	
30		Pakalniškiai 187	2	116.6		-	78	
31		Pakalniškiai 190	2	117.1		-	1314	spavin
32		Pakalniškiai 192.1	4	114.9		15.2	5–6	
33		Pakalniškiai 192.2	1	150.9		-	1.5+	

**Table 3 animals-12-01549-t003:** Horses from Kernavė medieval town and castles studied.

No.	Site	Number of Measured Bones	WH Min	WH Max	WH Mean	SD	Number of Measured Metacarpals	MC SD/GL × 100 Min	MC SD/GL × 100 Max	MC SD/GL × 100 Mean	MC SD/GL × 100 SD
1	Kernavė town	25	108.7	143.4	128.2	9.1	11	13.1	16.2	14.6	0.9
2	Vilnius Lower castle	9	119.0	155.6	128.1	11.0	5	14.4	15.7	15.2	0.5
3	Memelburg, Order castle	8	115.8	148.5	132.8	12.4	3	15.1	15.4	15.3	0.1

## Data Availability

Not applicable.

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
