# Peer review of "Horses in Lithuania in the Late Roman–Medieval Period (3rd–14th C AD) Burial Sites: Updates on Size, Age and Dating"

_animals, 2022, doi:10.3390/ani12121549_

Round 1

Reviewer 1 Report

The article ‘Horses in Lithuania in Late Roman-Medieval period (3rd-14th c. AD) burial sites: updates on size, age, and dating’ presents an osteological and osteometric investigation of Lithuanian horses from the 3rd to the 14th c. AD.

Overall, it is a good paper that is worth publishing because of the importance of burying horses in this region during the Late Roman period and beyond. Also, it contributes to the discussion of some trends about the introduction and movement of horses along the history that have been suggested for other areas of Europe. However, although the data are carefully presented, there are some things that should be reviewed before accepting it for publication. I would be willing to review a revised version if the following weaknesses pointed are improved:

Material and methods section:

- The authors should explain the methodology applied to differentiate between horses, donkeys and mules, morphologically very similar species.

- The authors make no mention of sex determination of horse remains. Although do not exist great sexual dimorphism between horse males and females, this is an important point and there are some external elements that can be used to separate them. For example, well-developed canines are present in all male equids but are rarely present in mares.

- Some explanations about the reasons for selecting only 13 horse remains for radiocarbon dating should be appreciated. Why were horse samples from other periods not radiocarbon dated?

- The authors should explain the reasons to include in the study three sites (Kernavé, Vilnius Lower castle and Memerlburg castle) in which horse remains were not recovered in burials.

Discussion section:

- The discussion section is a bit confusing and hard to read in its current format. Most of the arguments presented here are based on the fact that the Viking period horses were like the local horses, assuming that small horses were locals and the large ones non-locals (lines 530, 540 for example). But no data is presented about the size and robusticity of pre-Roman and/or early Roman horses (locals?). Only some data about Germanic and Gaul horses are presented (lines 445-449). Maybe it would be necessary to present some data about the size of pre-late Roman Lithuanian horses to provide robustness and solidity to the arguments presented in the discussion. If this is not possible, the paragraphs that attempt to describe the size of the local type of horses (lines 432-463) should be rewritten.

- The authors should explain (or at least present some hypothesis) the different dynamic of robusticity observed between metacarpals and metatarsals (Figure 7, lines 595-600).

Specific comments:

- A clarification on the number of horse remains analysed in this study would be appreciated. Lines 13 and 109 refers to over 200 individuals and line 117 refers to 400 horse skeletal remains. NISP and NMI should be clearly presented.

- Present a combined Figure 2 with more photographs from other horse burials explained in the text would be appreciated.

- Does paragraph from line 319 to line 322 refer to table 3? Please add reference to the corresponding table.

- Figure 9 is very explanatory. However, to be more informative, it would be important to present the external data also by periods, as the Lithuanian data.

Author Response

We'd like to thank the Reviewer for his very valuable comments.

Here's our response to Reviewer 1 comments:

1.  The authors should explain the methodology applied to differentiate between horses, donkeys and mules, morphologically very similar species.

1. Done. We described methodology in the text: In order to differentiate between horses (Equus caballus), donkeys (Equus asinus) and mules, the methodology described by Johnstone [12] was followed.

2. The authors make no mention of sex determination of horse remains. Although do not exist great sexual dimorphism between horse males and females, this is an important point and there are some external elements that can be used to separate them. For example, well-developed canines are present in all male equids but are rarely present in mares.

2. Explained (horse skeletons have been sexed based on the presence of canine teeth in male horses over 4–5 years of age. However, in many cases, sex of the animal could not be determined due to missing bones. Data on animal sex was added to the tables.)

3. Some explanations about the reasons for selecting only 13 horse remains for radiocarbon dating should be appreciated. Why were horse samples from other periods not radiocarbon dated?

3. Explained in the text. For 14C dating were selected all horses analysed for 87Sr/86Sr.

4. The authors should explain the reasons to include in the study three sites (Kernavé, Vilnius Lower castle and Memerlburg castle) in which horse remains were not recovered in burials.

4. Explained in the text. (We have included horses from the settlement sites in order to give as objective a picture of medieval horses as possible, based not only on the individuals from ordinary burial sites, but also from contexts related to the higher social strata (Vilnius, Kernavė) and the Order.) 

5. The discussion section is a bit confusing and hard to read in its current format. Most of the arguments presented here are based on the fact that the Viking period horses were like the local horses, assuming that small horses were locals and the large ones non-locals (lines 530, 540 for example). But no data is presented about the size and robusticity of pre-Roman and/or early Roman horses (locals?). Only some data about Germanic and Gaul horses are presented (lines 445-449). Maybe it would be necessary to present some data about the size of pre-late Roman Lithuanian horses to provide robustness and solidity to the arguments presented in the discussion. If this is not possible, the paragraphs that attempt to describe the size of the local type of horses (lines 432-463) should be rewritten.

5. Situation was described in lines 434-438. We have size (122 cm) of one early-Roman horse.

6. The authors should explain (or at least present some hypothesis) the different dynamic of robusticity observed between metacarpals and metatarsals (Figure 7, lines 595-600).

6. Added to the text:  Unfortunately, we still have no plausible explanation for the different dynamics of robusticity, observed between metacarpals and metatarsals. Perhaps we can tentatively speculate that the robusticity of the Viking period horses may reflect the specificity of the local horse type, while in the Medieval period, the influence of non-local horses with a different build may have become more prominent.

7. A clarification on the number of horse remains analysed in this study would be appreciated. Lines 13 and 109 refers to over 200 individuals and line 117 refers to 400 horse skeletal remains. NISP and NMI should be clearly presented.

7. Explained in the text: Remains of ca 400 horses were analysed, however due to poor preservation only over 200 individuals were suitable for further size and age analysis

8. Present a combined Figure 2 with more photographs from other horse burials explained in the text would be appreciated.

8. One more picture (Pagrybis 207 horse burial) was added to Figure 2.

9. Does paragraph from line 319 to line 322 refer to table 3? Please add reference to the corresponding table.

9. It related to Table 2, reference added.

10. Figure 9 is very explanatory. However, to be more informative, it would be important to present the external data also by periods, as the Lithuanian data.

10. We tried to change figure creating different colors or symbols for external data. However, picture looks very confusing and unclear then, therefore we decided do not change it avoiding the overload of illustration. Moreover, period for external data was mentioned near every external dot. We just added period for Sambia material.

Reviewer 2 Report

This paper presents new data on ancient horse remains from museums in Lithuania. It is well-written and easy to follow the information. The main thing that should be revised is: What is the main question that is being addressed with this data? I agree that the information is good to publish but what would you like to know or test with it? Why is it interesting to look at size change over time here? Why is it so important to study horse-human burials in this place over time? How do the analyses of these attributes fit together to investigate cultural change over time? Adding a hypothesis or main thesis statement to the first paragraph and the abstract will help to clarify the main goal of this paper. For example, by the time the results are presented, it is not clear how to evaluate the size information. Is the average height small or large relative to other places? There is a good hypothesis presented in the results section. These could be moved to the beginning of the paper to let the reader know what to look for in the results. This would help with understanding the conclusion as well. The authors state that the view of horses in Lithuania has been corrected by their findings…but what was the previous view that was incorrect? What does this tell us about broader human history with horses?

Reorganization of the order of the paper sections would also help to build up the argument with a background and a methods section defined, depending on the main question that is asked. Also, if calvary, combat tactics, and social composition are not sections of the discussion, perhaps they should not be brought up in the intro and conclusion as main subjects related to this data.

The charts and tables are well made. For figure 3, is there a reason to order the dates from top to bottom in this way? Could you re-organize them by site proximity or some other factor? Also, in section 4.2, please state which test of significance was used in line 411. Figure 5 may also be easier to read if presented as %bar graphs for the % total individuals in each height category for each period. In Figure 10, this would be easier to evaluate if the age groups were presented for each period, as in, put the periods on the x-axis. State the type of significance test.

Please reference the dates of these periods as many seem country-specific such as the Migration period. There are also some minor English editing errors that the authors can revise but these are minimal. One is to add “the” before many of the period names, such as “the” Migration period. Also, add a definition for “spavin”.

Author Response

We'd like to than to Reviewer for his valuable comments.

Here's our response to Reviewer 2 comments:

1. What is the main question that is being addressed with this data? I agree that the information is good to publish but what would you like to know or test with it? Why is it interesting to look at size change over time here? Why is it so important to study horse-human burials in this place over time? How do the analyses of these attributes fit together to investigate cultural change over time? Adding a hypothesis or main thesis statement to the first paragraph and the abstract will help to clarify the main goal of this paper. For example, by the time the results are presented, it is not clear how to evaluate the size information. Is the average height small or large relative to other places? There is a good hypothesis presented in the results section. These could be moved to the beginning of the paper to let the reader know what to look for in the results. This would help with understanding the conclusion as well. The authors state that the view of horses in Lithuania has been corrected by their findings…but what was the previous view that was incorrect? What does this tell us about broader human history with horses? Reorganization of the order of the paper sections would also help to build up the argument with a background and a methods section defined, depending on the main question that is asked. Also, if calvary, combat tactics, and social composition are not sections of the discussion, perhaps they should not be brought up in the intro and conclusion as main subjects related to this data.

1. We expanded the abstract, main text and the conclusions, changes noted in the text.

2. The charts and tables are well made. For figure 3, is there a reason to order the dates from top to bottom in this way? Could you re-organize them by site proximity or some other factor?

2. We have considered three ways of arrangement of the dates, i.e. chronological, site proximity, and alphabetic order. The reasons we chose the latter are following: 1) The dates do not reflect a single dated event or chronological horizon, therefore chronological order would break distribution of burials within individual sites; 2) Chronological analysis of burial customs change in 2th-7th C. AD was not among the key tasks of the paper; 3) The alphabetic order best allows the reader to distinguish the needed dates; 4) arrangement by site proximity would not correspond to cultural attributions of the sites and would be also artificial as alphabetic is; 5) Alphabetic arrangement best corresponds to organization of the text (site description) and the tables (1 and 2).

3. Also, in section 4.2, please state which test of significance was used in line 411.

3. We used t-test, it was added to the section Material and methods.

 4. Figure 5 may also be easier to read if presented as %bar graphs for the % total individuals in each height category for each period.

4. Figure was changed.

5. In Figure 10, this would be easier to evaluate if the age groups were presented for each period, as in, put the periods on the x-axis. State the type of significance test.

5. We have decided don't change Figure 10 for several reasons. In the illustration, we wanted to demonstrate the differences in the proportions of age groups over the time, rather than the age structure of the horses in each period, and also we tried to make Figure 5 and Figure 10 the same design. 

6. Please reference the dates of these periods as many seem country-specific such as the Migration period. There are also some minor English editing errors that the authors can revise but these are minimal. One is to add “the” before many of the period names, such as “the” Migration period. Also, add a definition for “spavin”.

6. Dates of the periods are described in the text (see chapter of Material and methods), “the” was added before the period names.

Reviewer 3 Report

This is a high quality paper that presents a large volume of empirical data regarding stature, robusticity and dating of horses. This information is very well contextualized in relation to the archaeological sequence and useful observations are made. The authors avoid over-interpretation whilst providing very detailed discussion of what the data might mean. They are well aware of the literature and make suitable comparisons to other regions. Methods are well described and the illustrations and graphs are of high quality.

There is little to criticise. I would note that in line 344, when referring to trends in slenderness, the word 'significantly' is used without a test stat. Such stats are used for other criteria elsewhere in the paper, so perhaps there should be a stat used here (and word significantly used or not in accordance with that stat). 

Line 35: I think 'burnt' should probably be 'cremated'.

Author Response

We'd like to thank the Reviewer for his valuable comments.

Here's our response:

1. I would note that in line 344, when referring to trends in slenderness, the word 'significantly' is used without a test stat. Such stats are used for other criteria elsewhere in the paper, so perhaps there should be a stat used here (and word significantly used or not in accordance with that stat). 

 1. A stat was done, and added to the text

2. Line 35: I think 'burnt' should probably be 'cremated'.

2. Changed.

Reviewer 4 Report

I am very honoured to evaluate this work. The study reconsidered its broad assessment of the Late-Roman-Medieval period (3-14 centuries) in Lithuania. While this period makes a valuable contribution to the knowledge on horse burial, it shares important information about the structural features of these horses. This study contains valuable and original information.

In order to contribute to the study and to expand the scope of the evaluations, it will be of great benefit to consider some of the issues I have mentioned below.

1-While the study provides valuable information about Roman period and medieval horses, adding a comparison of Byzantine period horses will contribute in two different dimensions.

- Both the comparison of visual morphological characters within the same period and the effect on the withers height, if any, on the importation of horses into the country will be better understood. For this reason, considering the following article, opening a small paragraph and interpreting it will contribute.

Onar, V., Pazvant, G., Pasicka, E., Armutak, A., Alpak, H. (2015): Byzantine Horse Skeletons of Theodosius Harbour: 2. Withers height estimation. Revue de Médecine Vétérinaire 166 (1-2): 30-42.

2-Kiesewalter multipliers (Kiesewalter 1888, May 1985) were used to estimate the withers height in the study. These multipliers are widely used in zooarchaeological studies. However, considering that these formulations are adapted from assembled skeletons, it can be seen how much the change is according to the new multipliers. It will be useful to add the estimation made with metacarpal multipliers, which includes structures such as skin, muscle, joint space, with a sentence. There is no mistake in the evaluation made by the authors, and it is most natural for them to prefer the widespread evaluation. I think that it will be possible to see this change only in Lithuanian horses for the purpose of contributing to the study. For this purpose, I suggest you take a look at this post.

Onar, V., Kahvecioğlu, K.O., Olğun-Erdikmen, D., Alpak, H., Parkan-Yaramış, Ç., The estimation of withers height of ancient horse: New estimation formulations by using the metacarpal measurements of living horse. Revue Méd. Vét. 2018, 169(7-9): 157-165.

3- It is thought that Brauner metacarpal and metatarsal slenderness index scales are used in the evaluation made for metapodial slenderness indices. Metacarpal slenderness index is given in Tables 1 and 2 and metacarpal slenderness index is given in the text. I recommend adding the following two references to be cited as sources for these two indices.

De Grossi Mazzorin, J., A. Riedel, and A. Tagliacozzo. 1998. Horse remains in Italy from the Eneolithic to the Roman period. In Proceedings of the XIII International Congress of the UISPP, C. Peretto and C. Giunchi, eds. Forlí, Italy: Abaco, v. 6(1), 87–92.

Brauner metacarpal slenderness index scale

Metacarpal

Mazzorin et al. 1998

Very slender legged

less than 13.5

Slender legged

13.6-14.5

Slightly slender legged

14.6-15.5

Medium slender legged

15.6-16.5

Slightly massive legged

16.6-17.5

5

Massive legged

more than 17.5

6

Udrescu, M., Bejenaru, L., Hrişcu, C. 1999. Introducere în Arheozoologie. Iaşi, Romania. Pp:100-107.

Metatarsal

 Udrescu et al. 1999

Slender legged

less than 12.0

semi-slender legged

12.0-12.7

semi-massive legged

12.8-13.6

Massive legged

more than 13.7

4-The horse skull shown in Figure-6 is quite well preserved. There is a mention of "a fracture in the forehead region of the skull (frontal bone), probably caused by the blunt end of the ax". In order to eliminate the suspicion of whether the fracture is a taphonomic change or a tomb collapse, it will be useful to take a radiographic image and evaluate it and add it. This will clear the doubts about this issue.

5-Do skull of horses have pathologies (mouth area or other areas)? This may provide us with important information about their use and maintenance conditions (eg bit damage etc.).

6- It is appropriate to use AD in the entire article. For example: Not 7th C AD, true: 7th AD.

The study will have a much richer knowledge with the minor revisions mentioned above. I appreciate the detailed work of the authors. The study is well organized and provides detailed information on the archaeological remains of Lithuanian horses. I recommend accepting it after the specified minor revision.

Best Regards

Author Response

We'd like to thank the Reviewer for his very valuable comments.

Here's our response to Reviewer 4:

1. While the study provides valuable information about Roman period and medieval horses, adding a comparison of Byzantine period horses will contribute in two different dimensions. Both the comparison of visual morphological characters within the same period and the effect on the withers height, if any, on the importation of horses into the country will be better understood. For this reason, considering the following article, opening a small paragraph and interpreting it will contribute.

Onar, V., Pazvant, G., Pasicka, E., Armutak, A., Alpak, H. (2015): Byzantine Horse Skeletons of Theodosius Harbour: 2. Withers height estimation. Revue de Médecine Vétérinaire 166 (1-2): 30-42.

1. We included this paper to our manuscript.

2. Kiesewalter multipliers (Kiesewalter 1888, May 1985) were used to estimate the withers height in the study. These multipliers are widely used in zooarchaeological studies. However, considering that these formulations are adapted from assembled skeletons, it can be seen how much the change is according to the new multipliers. It will be useful to add the estimation made with metacarpal multipliers, which includes structures such as skin, muscle, joint space, with a sentence. There is no mistake in the evaluation made by the authors, and it is most natural for them to prefer the widespread evaluation. I think that it will be possible to see this change only in Lithuanian horses for the purpose of contributing to the study. For this purpose, I suggest you take a look at this post.

Onar, V., Kahvecioğlu, K.O., Olğun-Erdikmen, D., Alpak, H., Parkan-Yaramış, Ç., The estimation of withers height of ancient horse: New estimation formulations by using the metacarpal measurements of living horse. Revue Méd. Vét. 2018, 169(7-9): 157-165.

2. Thank you for your suggestion. However, in this work, we already present a large amount of data from different periods, settlements and countries. We have used a generally accepted methodology to calculate the withers height of horses, which allows to compare data from different authors and countries. We believe that this is sufficient, and afraid, that another methodology for calculating height would add more confusion and potentially overload the article.

3. It is thought that Brauner metacarpal and metatarsal slenderness index scales are used in the evaluation made for metapodial slenderness indices. Metacarpal slenderness index is given in Tables 1 and 2 and metacarpal slenderness index is given in the text. I recommend adding the following two references to be cited as sources for these two indices.

De Grossi Mazzorin, J., A. Riedel, and A. Tagliacozzo. 1998. Horse remains in Italy from the Eneolithic to the Roman period. In Proceedings of the XIII International Congress of the UISPP, C. Peretto and C. Giunchi, eds. Forlí, Italy: Abaco, v. 6(1), 87–92.

Brauner metacarpal slenderness index scale

Metacarpal

Mazzorin et al. 1998

Very slender legged

less than 13.5

Slender legged

13.6-14.5

Slightly slender legged

14.6-15.5

Medium slender legged

15.6-16.5

Slightly massive legged

16.6-17.5

5

Massive legged

more than 17.5

6

Udrescu, M., Bejenaru, L., Hrişcu, C. 1999. Introducere în Arheozoologie. Iaşi, Romania. Pp:100-107.

Metatarsal

 Udrescu et al. 1999

Slender legged

less than 12.0

semi-slender legged

12.0-12.7

semi-massive legged

12.8-13.6

Massive legged

more than 13.7

3. Yes, Brauner slenderness index scales were used, we added reference to the text.

4. The horse skull shown in Figure-6 is quite well preserved. There is a mention of "a fracture in the forehead region of the skull (frontal bone), probably caused by the blunt end of the ax". In order to eliminate the suspicion of whether the fracture is a taphonomic change or a tomb collapse, it will be useful to take a radiographic image and evaluate it and add it. This will clear the doubts about this issue.

4. Thank you for the great idea, it’s worthful to use in our further studies related to horse pathologies and burial practices. However, in this case we found visual analysis and context better than x-ray to state the fracture is related to horse killing and is not result taphonomic change. Our arguments were stated in lines 517-518. However, as regards the tool used, we are only making a cautious assumption which needs to be verified experimentally.

5. Do skull of horses have pathologies (mouth area or other areas)? This may provide us with important information about their use and maintenance conditions (eg bit damage etc.).

5. We add additional data about this cranium to the text. No pathologies were observed, however horse still had P1 teeth, and bit damage was observed on lower premolars.

6. It is appropriate to use AD in the entire article. For example: Not 7th C AD, true: 7th AD.

6. AD was added where it was missed. Formulation “7th C AD” was checked and approved by native English language translator, and was used also in other papers, therefore we left if.

Round 2

Reviewer 1 Report

Accept in present form.